# Generalization error bounds for iterative learning algorithms with bounded updates

## Abstract

This paper explores the generalization characteristics of iterative learning algorithms with bounded updates for non-convex loss functions, employing information-theoretic techniques. Our key contribution is a novel bound for the generalization error of these algorithms with bounded updates. Our approach introduces two main novelties: 1) we reformulate the mutual information as the uncertainty of updates, providing a new perspective, and 2) instead of using the chaining rule of mutual information, we employ a variance decomposition technique to decompose information across iterations, allowing for a simpler surrogate process. We analyze our generalization bound under various settings and demonstrate improved bounds. Ultimately, our work takes a further step for developing practical generalization theories.

## 1 Introduction

The majority of machine learning techniques utilize the empirical risk minimization framework. Within this framework, the optimization objective is to minimize empirical risk, which is the average risk over a finite set of training samples. In practice, the value of interest is the population risk, representing the expected risk across a population. Generalization error is the difference between the optimization objective (empirical risk) and the value of interest (population risk). The prevalence of machine learning techniques makes it essential to comprehend generalization error.

Previous studies (Russo & Zou, 2016; 2019; Xu & Raginsky, 2017) have established a relationship between mutual information, $I(W; S_n)$, and the generalization error, where $S_n$ is a set containing $n$ samples from a distribution $\mu$, erving as the algorithm's input, and $W$ represents the model's weights after training, serving as the algorithm's output. Information-theoretic tools are well-suited for analyzing iterative learning algorithms, as the chain rule of mutual information allows for a simple decomposition $I(W, S_n)$ across iterations (i.e. $I(W_T; S_n) \leq I(W_1, \cdots W_T; S_n) \leq \sum_{t=1}^{T} I(W_t; S_n | W_{t-1})$). Leveraging this technique, Xu & Raginsky (2017) studies the generalization properties of stochastic gradient Langevin dynamics (SGLD). SGLD can be considered as introducing noise to the SGD in each update step.

Since most commonly used algorithms in practice, such as SGD and Adam (Kingma & Ba, 2014), do not incorporate noise injection during the update process, recent research efforts are focused on integrating information-theoretic methods into these iterative algorithms without added noise. The challenge in this approach is that the value of $I(W_t; S_n | W_{t-1})$ will become infinite when $W_T$ is determined by $S_n$ and $W_{t-1}$. A potential solution involves utilizing surrogate processes (Negrea et al., 2020; Sefidgaran et al., 2022). Neu et al. (2021) derives generalization bounds for SGD by employing a "virtual SGLD" surrogate process, in which noise is introduced during each update step of (S)GD. Their generalization bound consists of two components: the generalization bound for the surrogate process and the bound for the difference between the generalization errors of the surrogate and original processes.

This paper examines the mutual information $I(S_n, W)$ from alternative perspectives and reformulates the mutual information to relate to the uncertainty of the update. The uncertainty of the update refers to how the update will vary for different datasets $S_n \sim \mu^{\otimes n}$. Instead of applying the chaining rule of mutual

information, we use a variance decomposition method to decompose information across iterations. From this perspective, we establish the generalization bound for general iterative algorithms with bounded updates by employing a surrogate process that adds noise exclusively to the original process's final update.

We analyze our generalization bound in different situation. Our work achieve better vanishing rate guarantee than previous work Neu et al. (2021). We also investigate the gap between our theoretical framework and practical applications by analyzing the previous discovery of the scaling behavior in large language models. Our model shed light on developing practically useful generalization theories.

The contributions of our work can be summarized as following:

- This paper offers a novel viewpoint for analyzing the mutual information $I(W, S_n)$ by focusing on the uncertainty of updates.

- A new generalization bound, derived from an information-theoretic approach, is presented. This bound is applicable to iterative learning algorithms with bounded updates.

- We investigate the generalization behavior of various types of bounded update, iterative learning algorithms.

## 2 Related works

Existing works on generalization theory can be roughly divided into two categories: function space based method, and the learning algorithm based method. The function space based method study the generalization behavior based on the complexity of function space. Many methods for measuring the complexity of the function space have been proposed, e.g., VC dimension (Vapnik & Chervonenkis, 2015), Rademacher Complexity (Bartlett & Mendelson, 2002) and covering number (Shalev-Shwartz & Ben-David, 2014). These works fail in being applied to overparameters models, where the number of parameters is larger than the number of data samples. Because the function space is too large to deliver a trivial result (Zhang et al., 2021) in this case. To overcome this problem, recent works want to leverage the properties of learning algorithm to analyzing the generalization behavior. The most popular methods are stability of algorithm (Hardt et al., 2016) and information-theoretic analysis (Xu & Raginsky, 2017; Russo & Zou, 2016). Among them, the stability of algorithm (Bousquet & Elisseeff, 2002) measures how one sample change of training data impacts the model weights finally learned, and the information theory (Russo & Zou, 2016; 2019; Xu & Raginsky, 2017) based generalization bounds rely on the mutual information of the input (training data) and output (weights after training) of the learning algorithm. Although the both the stability method and information theoretic method are general, obtaining the generalization bound for practical learning algorithms is non-trival. Most of the stability-based generalization bounds focus on SGD (Hardt et al., 2016; Bassily et al., 2020; Nikolakakis et al., 2022). Applying the stability-based method outside SGD is very complex and non-trival (Nguyen et al., 2022; Ramezani et al., 2018). Most information-theoretic generalization bounds are applied for Stochastic Gradient Langevin Dynamics(SGLD), e.g., SGD with noise injected in each step of parameters updating (Pensia et al., 2018; Negrea et al., 2019; Haghifam et al., 2020; Negrea et al., 2019; Haghifam et al., 2020). Neu et al. (2021) extends the information-theoretic generalization bounds to SGD by leveraging surrogate process. **Our work advances the field by extending the information-theoretic based method to learning algorithms beyond SGD in a simple way.** This represents a significant step towards developing practically useful generalization theories.

## 3 Preliminary

Let $P, Q$ be probability measures on a measurable space. When $Q \ll P$, meaning $Q$ is absolutely continuous with respect to $P$, $\frac{dQ}{dP}$ represents the Radon-Nikodym derivative of $Q$ concerning $P$. The relative entropy (KL divergence) is calculated as $\mathrm{KL}(P\|Q) = \int_x dP(x) \log\left(\frac{dP}{dQ}(x)\right)$. The distribution of variable $x$ is denoted as $\mathbb{P}(x)$ or $\mathbb{P}_x$. The product distribution between two variables $x, y$ is denoted as $\mathbb{P}(x) \otimes \mathbb{P}(y)$. The mutual information between two variables $x, y$ is calculated as $I(x; y) = \mathrm{KL}(\mathbb{P}(x, y)\|\mathbb{P}(x) \otimes \mathbb{P}(y))$. We use $\|\cdot\|$ to denote the Euclidean norm. And we denote $\{1, \cdots, k\}$ as $[k]$.

We consider the data distribution $\mu$. The data $Z$ is sampled from $\mu$ and resides in the space $\mathcal{Z}$. The training dataset is represented as $S_n \sim \mu^{\otimes n}$. The learning algorithms is denoted as $\mathcal{A}$ which takes $S_n$ as input and outputs weights for parameterized. The weights are denoted as $W \in \mathbb{W}$, with a dimension of $d$. The performance and behavior of these weights are evaluated using a loss function, represented as $f(W, Z) \in \mathbb{R}_+$. We assume $f(W, Z)$ is differentiable with respect to $W$. The gradient and the Hessian matrix of $f(W, Z)$ are denoted as $\nabla f(W, Z)$ and $\nabla^2 f(W, Z)$ respectively. the value of interest is population risk, which is calculated as

$$F_\mu(W) = \mathbb{E}_{z \sim \mu} f(W, z). \tag{1}$$

However, the population risk is often inaccessible. In the context of empirical risk minimization (ERM), the objective is to minimize the empirical risk. Given a data set $S_n = \{z_i\}_{i=1}^n \sim \mu^{\otimes n}$, the empirical risk is calculated as

$$F_{S_n}(W) = \frac{1}{n} \sum_{i=1}^n f(W, z_i). \tag{2}$$

The empirical risk is determined by averaging all samples in a dataset $S_n$. This paper primarily focuses on the generalization error, which represents the difference between empirical risk and population risk. The generalization error can be calculated as follows

$$gen(\mu, \mathbb{P}_{W|S_n}) = \mathbb{E}_{S_n \sim \mu^{\otimes n}, W \sim \mathbb{P}_{W|S_n}} \left[ F_{S_n}(W) - F_\mu(W) \right]. \tag{3}$$

The generalization error is calculated as the expectation concerning the randomness of data and the algorithm. In the learning problem, we iteratively update the weights of parameterized functions. We represent the weights at step $t$ as $W_t$. $W_t$ is acquired by adding the update value to the initial weights $W_0$, i.e., $W_t = W_{t-1} + U_t$. Typically, $U_t$ takes the form $U_t = \eta_t u_t$, where $\eta_t$ indicates the learning rate for the $t$-th step. We denote the accumulated update as $U^{(t)} \triangleq \sum_{t=1}^t U_t$. The initial weights are obtained by sampling from a specific distribution, i.e. $W_0 \sim \mathbb{P}(W_0)$. The final output of the $T$-steps algorithm is $W_T$. The variance of update is defined as:

$$\mathbb{V}_{\mu,n}(U^{(t)}|W_0) \triangleq \mathbb{E}_{W_0 \sim \mathbb{P}_{W_0}} \mathbb{E} \left[ \left\| U^{(t)} - \mathbb{E}U^{(t)} \right\|^2 |W_0 \right], \tag{4}$$

where the $\mathbb{E}U^{(t)}$ is taking the expection of all randomness of $U^{(t)}$, including the randomness caused by data sampling and the randomness of learning algorithm. Following the similar way, we define the covariance as

$$\mathbb{C}_{\mu,n}(U_i, U_j|W_0) \triangleq \mathbb{E}_{W_0 \sim \mathbb{P}_{W_0}} \mathbb{E} \left[ < \bar{U}_i, \bar{U}_j > |W_0 \right], \tag{5}$$

where $\bar{U}_i = U_i - \mathbb{E}U_i$. Without loss of ambiguity, we simplify $\mathbb{V}_{\mu,n}(U^{(t)}|W_0)$ as $\mathbb{V}(U^{(t)})$ and $\mathbb{C}_{\mu,n}(U_i, U_j|W_0)$ as $\mathbb{C}(U_i, U_j)$.

## 4 Generalization bound

Our primary result is a bound on the generalization error of the weights $W$ generated by a learning algorithm with bounded updates. We will initially analyze the generalization mutual information from the perspective of update uncertainty. Subsequently, we will provide a bound for the learning algorithm with bounded updates.

### 4.1 Generalization bounds with uncertainty of update

We begin by discussing the assumption used in our bound. The $R$-sub-Gaussian is defined as follows:

**Definition 4.1.** A random variable $X$ is $R$-sub-Gaussian if for every $\lambda \in \mathbb{R}$, the following inequality holds:

$$\mathbb{E}[\exp\left(\lambda(X - \mathbb{E}X)\right)] \leq \exp\left(\frac{\lambda^2 R^2}{2}\right) \tag{6}$$

*Remark* 4.2. If a variable $X \in \mathbb{R}$ and takes value in $[a, b]$, then the variable is $(b - a)/2$-sub-Guassian.

Based on the definition of $R$-sub-Guassian, our assumption is:

**Assumption 4.3.** Suppose $f(w, Z)$ is $R$-sub-Guassian with respect to $Z \sim \mu$ for every $w \in \mathcal{W}$.

With the $R$-sub-Guassian, we obtain the following generalization bound,

**Theorem 4.4.** *Under Assumption 4.3, the following bound holds:*

$$\left| gen(\mu, \mathbb{P}_{W|S_n}) \right| \leq \sqrt{\frac{2R^2}{n}[h(U^{(T)}|W_0) - h(U^{(T)}|W_0, S_n)]}. \tag{7}$$

This bound transfer the original $I(W; S_n)$ into the difference between two update entropy. The update entropy can be interprete as of measure the uncertainty. $h(U^{(T)}|W_0) - h(U^{(T)}|W_0, S_n)$ measures the contribution dataset $S_n$ to the update uncertainty. A low generalization bound can be obtained if the learning algorithm takes a similar update given different $S_n \sim \mu^{\otimes n}$.

We first consider the situation where $h(U^{(T)}|W_0, S_n) \geq 0$. In this case, we can simply omit $h(U^{(T)}|W_0, S_n)$ and we only need to derive a upper bound of $h(U^{(T)}|W_0)$.

**Theorem 4.5.** *Under Assumption 4.3, for high randomness learning algorithm, i.e. $h(U^{(T)}|W_0, S_n) \geq 0$, the generalization error of the final iteration satisfies*

$$\left| gen(\mu, \mathbb{P}_{W|S_n}) \right| \leq \sqrt{\frac{2\pi e R^2 \mathbb{V}(U^{(T)})}{n}}. \tag{8}$$

*Remark* 4.6. $h(U^{(T)}|W_0, S_n) \geq 0$ can be achieved if the learning algorithms have hign randomness. The high randomness can be obtain through 1) using small batch size 2) adding noise during the updates, like SGLD 3) or some other methods.

The generalization bound in Theorem 4.4 can not be calculated directly when $h(U^{(T)}|W_0, S_n) < 0$, because we don't know the distribution of $U^{(T)}$. The $h(U^{(T)}|W_0)$ and $h(U^{(T)}|W_0, S_n)$ can be extremely small when the algorithm has minimal randomness. A natural approach is to associate the update entropy with the Gaussian distribution entropy, which can be calculated directly. Consequently, we introduce a surrogate process for our analysis:

**surrogate process** We consider the surrogate update $\tilde{U}$ with noise added to the final update, i.e., $U_t = U_t$ when $t \neq T$ and $U_T = U_T + \epsilon$, where $\epsilon$ is a random noise. Here we consider $\epsilon \sim \mathcal{N}(0, \sigma^2 \mathbf{I})$. Then we have $\tilde{U}^{(T)} = U^{(T)} + \epsilon$.

Based on the surrogate process, we obtain the result:

**Theorem 4.7.** *Under Assumption 4.3, for any $\sigma$, the generalization error of the final iteration satisfies*

$$\left| gen(\mu, \mathbb{P}_{W|S_n}) \right| \leq \sqrt{\frac{R^2 \mathbb{V}(U^{(T)})}{n\sigma^2}} + \Delta_\sigma, \tag{9}$$

*where $\Delta_\sigma \triangleq |\mathbb{E}\left[(F_\mu(W_T) - F_\mu(W_T + \epsilon)) - (F_S(W_T) - F_S(W_T + \epsilon))\right]|$ and $\epsilon \sim \mathcal{N}(0, \sigma^2 \mathbf{I})$.*

*Remark* 4.8. Compared to Theorem 4.5, Theorem 4.7 employs the surrogate process and, as a results, this theorem is more general. We give a further analysis of the results of this Theorem from Pac-Bayes perspective in Appendix G to remove sub-Guassian assumption and obtain high probability bounds.

Theorem 4.5 and Theorem 4.7 establish a connection between the generalization error and the variance of the update. Based on this, the generalization analysis of bounded updates learning algorithm is given in the following section.

## 4.2 Generalization bounds for bounded updates learning algorithms

Building on the results from the previous section, we derive the bound for the bounded updates learning algorithm in this part. We provide the formal definition of the bounded updates as follows:

**Definition 4.9.** (Bounded updates) A learning algorithm is said to have bounded updates with respect to function $f(\cdot)$ and data distribution $\mu$, if for all $S_n \sim \mu^{\otimes n}$, there exists a constant $L$, such that $\|u_t\| \leq L$ for all $t \leq T$, when the learning algorithm is operated on $f(\cdot)$ and $S_n$.

**Comparison between bounded updates assumption and $L$-Lipschitz assumption**   The $L$-Lipschitz assumption is widely used to analyze the convergence or generalization behavior of learning algorithms. The $L$-Lipschitz condition requires that $\|\nabla f(w, Z)\| \leq L$ for all $w, Z$. These two assumptions, $L$-Lipschitz and bounded update, share some similarities. However, some fundamental differences exist: **1)** $L$-Lipschitz is a property of $f(\cdot)$, while the bounded updates is a joint behavior of the learning algorithm and $f(\cdot)$. It is possible to achieve a bounded updates behavior even when the function is not $L$-Lipschitz. **2)** The $L$-Lipschitz is a "global assumption," meaning that the assumption must be held for all $w$. On the other hand, the bounded updates assumption is a local assumption. This assumption is only required to be held for the weights encountered during the learning process.

Under the bounded updates assumption, we can obtain the result as follows:

**Theorem 4.10.** *If the learning algorithm has bounded updates on data distribution $\mu$ and loss function $f(\cdot)$, then we have*

$$\mathbb{V}(U^{(T)}) \leq \sum_{t=1}^{T} 4\eta_t^2 L^2 + 2L^2 \sum_{t=1}^{T} \eta_t \sum_{i=1}^{t-1} \eta_t \tag{10}$$

*then under Assumption 4.3, we have*

$$gen(\mu, \mathbb{P}_{W|S_n}) \leq \sqrt{\frac{R^2}{n\sigma^2} \left( \sum_{t=1}^{T} 4\eta_t^2 L^2 + 2L^2 \sum_{t=1}^{T} \eta_t \sum_{i=1}^{t-1} \eta_t \right)} + \Delta_\sigma. \tag{11}$$

*If the learning algorithms have high randomness, i.e. satisfying $h(U^{(T)}|W_0, S_n) \geq 0$, we have*

$$\left| gen(\mu, \mathbb{P}_{W|S_n}) \right| \leq \sqrt{\frac{2\pi e R^2}{n} \left( \sum_{t=1}^{T} 4\eta_t^2 L^2 + 2L^2 \sum_{t=1}^{T} \eta_t \sum_{i=1}^{t-1} \eta_t \right)}. \tag{12}$$

**Proof Schetch:**   The full proof is listed in Appendix D. Here, we give the proof schetch. **Step 1** We use the equation $\mathbb{V}(U^{(T)}) = \sum_{t=1}^{T} \mathbb{V}(U_t) + 2\sum_{t=1}^{T} \mathbb{C}(U^{(t-1)}, U_t)$ to decomposite the $\mathbb{V}(U^{(T)})$ to the information along the learning trajectory. **Step 2** Due to the bounded updates assumption, the $\mathbb{V}(U_t) \leq 4\eta_t^2 L^2$ and $\mathbb{V}(U^{(t)}) \leq L \sum_{i=1}^{t} \eta_t$. **Step 3** Combining the results above, we obtain the final bound.

**Technique Novelty:**   Most previous works employ the technique $I(W_T; S_n) \leq \sum_{t=1}^{T} I(W_t; S_n | W_{t-1})$ to decompose the information of the final weights into the information along the learning trajectory. This method fails in our case because we do not add noise at every update step along the learning trajectory. As a result, $I(W_t; S_n | W_{t-1})$ becomes large in this situation. To address this challenge, we utilize another commonly used technique: $\mathbb{V}(U(T)) = \sum_{t=1}^{T} \mathbb{V}(U_t) + 2\sum_{t=1}^{T} \mathbb{C}(U(t-1), U_t)$. This method is quite simple, but it is effective. We will analyze the effectiveness of our method by comparing it with Neu et al. (2021), which uses the technique $I(W_T; S_n) \leq \sum_{t=1}^{T} I(W_t; S_n | W_{t-1})$, in the following section.

## 5   Analysis

### 5.1   bounded updates learning algorithms

In this section, we will discussion about the bounded updates hehavior of commonly used algorithm.

**Proposition 5.1.** *Adam(Kingma & Ba, 2014), Adagrad(Duchi et al., 2011), RMSprop(Tieleman et al., 2012) are bounded updates with respect to all data distribution and function $f(\cdot)$ when $d = \mathcal{O}(1)$*

This proposition suggests that when setting the dimension $d$ as a constant, commonly used learning algorithms, such as Adam, Adagrad, and RMSprop, exhibit bounded updates. However, in real-world situations, we typically scale the model size based on the amount of data, which implies that $d$ will increase along with $n$. In this scenario, we do not have $d = \Theta(1)$.

Then, we consider the learning algorithm modified with update clip. The update rule of learning algorithm with update clip is $u_t = \min\{L, \|u_t'\|\}\frac{u_t'}{\|u_t'\|}$, where $u_t'$ is the update value of original learning algorithm without update clip.

**Proposition 5.2.** *All learning algorithms with update clip and SGD(M) with grad clip have bounded updates with respect to all data distribution and function $f(\cdot)$.*

*Proof.* For algorithms with update clip, we have $\|u_t\| = \min\{L, \|u_t'\|\}\frac{\|u_t'\|}{\|u_t'\|} \leq L$. For (S)GD, because $u_t'$ is gradient of a batch data, the grad clip is equal to update clip. $\square$

The gradient clipping technique is commonly employed in practice (Zhang et al., 2019; Qian et al., 2021). If a learning algorithm does not have a bounded update, it may be possible to incorporate an update clipping technique to ensure that it aligns with our theoretical framework.

### 5.2 $d$ dependence of $\Delta_\sigma$

We consider the situations where $\sigma$ is a small value. As our analysis concentrates on the asymptotic behavior of the generalization error when $n$ increases, we use the setting $\lim_{n\to\infty} \sigma = 0$. In this situation, $\sigma$ is a small value when a relatively large $n$ is adopted.

For $z \in \mathcal{Z}$, we have

$$
\begin{aligned}
\mathbb{E}[f(W_T, z) - f(W_T + \epsilon, z)] &\approx \mathbb{E}[< \nabla f(W_T, z), \epsilon >] + \frac{1}{2}\mathbb{E}[\epsilon^{\mathrm{T}}\nabla^2 f(W_T, z)\epsilon] \\
&= \frac{1}{2d}\mathbb{E}\|\epsilon\|^2 \mathbb{E}\,\mathrm{Tr}(\nabla^2 f(W_T, z)) = \frac{\sigma^2}{2}\mathbb{E}\,\mathrm{Tr}(\nabla^2 f(W_T, z))
\end{aligned}
\tag{13}
$$

The, according to the definition of $\Delta_\sigma$, we have $\Delta_\sigma \approx \frac{\sigma^2}{2}\left|\mathbb{E}\,\mathrm{Tr}\left(\nabla^2 F_\mu(W_T) - \nabla^2 F_{S_n}(W_T)\right)\right|$. Therefore, analyzing $d$ dependence of $\Delta_\sigma$ is equal to analyzing the $d$ dependence of $\mathrm{Tr}\left(\nabla^2 f(W_T, z)\right)$.

**Worst case: $\Delta_\sigma = \Theta(d\sigma^2)$.** We assume the $\beta$-smooth for function $f(w, z)$, then we have the upper bound $\mathbb{E}\left|\mathrm{Tr}(\nabla^2(W_T, z))\right| \leq d\beta$. The equal sign is taken when all the eignvalue of $\nabla^2 f(W_T, z))$ is $\beta$.

**Benign case:** The benign case is possible when the distribution of eigenvalues of the Hessian matrix exhibits a long tail. In this situation, most eigenvalues are close to 0, which implies that $\mathrm{Tr}(\nabla^2(W_T, z))$ remains stable when increasing $d$. The long tail distribution is commonly observed in neural networks (Ghorbani et al., 2019; Sagun et al., 2016; Zhou et al., 2022). We consider two cases in this context: **1)** $\Delta_\sigma = \Theta(\sigma^2/\eta)$: This case may be achieved by leveraging the inductive bias of training algorithm. Wu et al. (2022) finds that the SGD can only converge to $W_T$ where $\mathrm{Tr}(\nabla^2(W_T, z))$ is smaller than a specific value. The value is dimension independent but learning rate dependent ($\frac{1}{\eta}$). The similar learning rate dependent on maximum eigenvalue is also discovered by Cohen et al. (2021; 2022). **2)** $\Delta_\sigma = \Theta(\sigma^2)$. This case may be achieved if the learning algorithm explicitly decreases $\mathrm{Tr}(\nabla^2(W_T, z))$. The SAM learning algorithm (Foret et al., 2020) is specifically designed to reduce the sharpness (maximum eigenvalue of the Hessian matrix). Wen et al. (2022) find that the stochastic SAM minimizes $\mathrm{Tr}(\nabla^2(W_T, z))$.

### 5.3 Compared with Neu et al. (2021)

Neu et al. (2021) consider the surrogate process that $\tilde{U}_t = U_t + \epsilon_t$ for all $t \in [T]$, where $\epsilon_t \sim \mathcal{N}(0, \sigma_t\mathrm{I}_d)$. They obtain the generalization error bound,

$$
\left|gen(\mu, \mathbb{P}_{W|S_n})\right| = \mathcal{O}(\sqrt{\frac{R^2\eta^2 T}{n}(dT + \frac{1}{b\sigma^2})} + \Delta_{\sigma_{1:T}}),
\tag{14}
$$

where $b$ denotes batch size and $\sigma_{1:T} = \sqrt{\sigma_1^2 + \cdots + \sigma_T^2}$.

Table 1: **Compared with information-theoretic method.** We give the asymptotic analysis when increasing $n$ under $T\eta = \Theta(1)$ for various scenarios, with $b$ denoting the batch size. ($\triangle$) stands for $h(U^{(T)}|W_0, S_n) < 0$, and ($\star$) is short for $h(U^{(T)}|W_0, S_n) \geq 0$.

| Settings | | Ours | | Neu et al. (2021) | |
|---|---|---|---|---|---|
| $d$ | $\Delta_\sigma$ | ($\triangle$) | ($\star$) | $b = \mathcal{O}(1)$ | $b = \mathcal{O}(\sqrt{n})$ |
| $\Theta(1)$ | $\Theta(d\sigma^2)$, $\Theta(\sigma^2/\eta)$ ,$\Theta(\sigma^2)$ | $\mathcal{O}(1/n^{\frac{1}{3}})$ | $\mathcal{O}(1/n^{\frac{1}{2}})$ | $\mathcal{O}(1/n^{\frac{1}{3}})$ | $\mathcal{O}(1/n^{\frac{1}{2}})$ |
| $\Theta(n)$ | $\Theta(d\sigma^2)$ | $\mathcal{O}(1)$ | | | |
| | $\Theta(\sigma^2/\eta)$ | $\mathcal{O}(1/n^{\frac{1}{3}})$ | $\mathcal{O}(1/n^{\frac{1}{2}})$ | $\mathcal{O}(1)$ | $\mathcal{O}(1)$ |
| | $\Theta(\sigma^2)$ | $\mathcal{O}(1/n^{\frac{1}{3}})$ | | | |

We consider two settings of $d$ in this analysis. The first one is the underparameterized regime, where $d = \Theta(1)$. In this regime, as we increase $n$ to a large value, $n$ will be significantly larger than $d$. The second setting is the overparameterized regime, where $d = \Theta(n)$. In this case, the ratio between $d$ and $n$ remains nearly constant as we increase $n$. This setting is commonly employed in Large Language Models (Muennighoff et al., 2023; Hoffmann et al., 2022) when scaling $n$. Table 1 examines the behavior of the generalization bound under different $d$ values and various cases of $\Delta_\sigma$. In this analysis, we fix $\eta T = \Theta(1)$.

**Last iteration noise v.s. whole process noise**  Our work and Neu et al. (2021) both utilize surrogate processes for analysis. The main difference lies in the surrogate process, where our approach adds noise only to the final iteration, while Neu et al. (2021) adds noise throughout the entire process. **Our bound is better for analysis** because our bounds only require taking infinity with respect to one variable $\sigma$, whereas the bound of Neu et al. (2021) needs to consider infinity with respect to $T$ variables, $\sigma_1, \cdots \sigma_T$. **Our method exhibits weaker dependence on $T$.** The $\Delta_\sigma$ used in our bound does not have a clear dependence on $T$, while the $\Delta_{\sigma_{1:T}}$ will increase with respect to $T$.

**Applies to general learning algorithms.**  Our bound don't leverage any specific knowledge about particular learning algorithms, while the main Theorem of Neu et al. (2021) only applied to (S)GD. Although the methods of Neu et al. (2021) is general which makes it possible to apply to other learning algorithms, it is untrival to do this. More information can be found in the Section "5. Extension" in Neu et al. (2021).

### 5.4  Compared with stability based method

Table 2: **Compared with stability-based works.** We consider the case where $\eta_t = \frac{1}{t}$ and $T = \mathcal{O}(n)$ to calculate the rate. Because stability-based works consider the stochastic optimizers with batch size 1, we choose our results with $h(U^{(T)}|W_0, S_n) \geq 0$ for fair comparison. The conclusion is that 1) Our method has weaker assumption on function $f(\cdot)$, and 2) Our bound achieve a better rate on non-convex function.

| Paper | Position (in ori paper) | Assumption | | | Learning algorithm | Rate |
|---|---|---|---|---|---|---|
| Hardt et al. (2016) | Thm 3.8 | Lipschtz | $\beta$-smooth | | SGD | $\mathcal{O}(1/n^{\frac{1}{\beta+1}})$ |
| Ramezani et al. (2018) | Thm 5 | Lipschtz | $\beta$-smooth | | SGDM | $\mathcal{O}(1/\log n)$ [1] |
| Lei & Ying (2020) | Thm 3 | nonnegative | convex | $\beta$-smooth | SGD | |
| Nguyen et al. (2022) | Thm 4 | bounded $f(\cdot)$ | Lipschtz | $\beta$-smooth | Adam, Adagrad | $\mathcal{O}(e^n/n)$ [2] |
| Ours | Thm 4.10 | sub-Guassian | | | Bounded update | $\mathcal{O}(\log n/\sqrt{n})$ |

Table 2 summaries some recent stability-based studies on different learning algorithms. Our methods have the following advantages:

- **Weaker assumptions.** Most stability-based works (Hardt et al., 2016; Ramezani et al., 2018; Nguyen et al., 2022) require Lipschitz and smooth assumption. Lei & Ying (2020) removes the

---

[1] We set $t_d$ in this bounds as $\mathcal{O}(T)$.
[2] Corollary 1 in Nguyen et al. (2022)

Lipschitz assumption, but the convex assumption is required. Our methods only require to be $f(\cdot)$ sub-Guassian.

- **Better results in non-convex situation.** Obviously, our methods have a better result than Nguyen et al. (2022); Ramezani et al. (2018) from Table 2 under the setting $\eta = \frac{1}{t}$ and $T = \mathcal{O}(n)$. As for the Hardt et al. (2016), our bound is better if $\beta > 1$, which is hold in many situations (Cohen et al., 2021; Ghorbani et al., 2019; Zhou et al., 2022).

*Remark* 5.3. We don't compare the results with Lei & Ying (2020) because 1) it relies on the convex assumption and 2) its studies don't include the results with learning rate setting $\eta_t = \frac{1}{t}$. Haghifam et al. (2023) argues that all the information-theoretic methods will be worse than stability-based work in convex case. We leave achieving better results in convex case using information-theoretic methods as future work (Detail discussion on Section 6).

### 5.5 Comparison between Theoretic and practice

**Integrating the knowledge of learning trajectory** Incorporating information from the learning trajectory is crucial for gaining a deeper understanding of generalization behavior. Fu et al. (2023) employs learning trajectory data to establish a better generalization bound for SGD. Additionally, using learning trajectory information could potentially enhance the bounds of iterative learning algorithms with bounded updates.

## 6 Limitation

Haghifam et al. (2023) analyzes the behavior of information-theoretic generalization bounds and stability-based generalization bounds, finding that all information-theoretic-based generalization bounds do not achieve a min-max rate comparable to stability-based works in stochastic convex optimization problems. **Our work cannot overcome this limitation for the following reasons**: 1) Unlike stability-based work, information-theoretic methods, including our work, cannot directly leverage convex information. This makes the information-theoretic methods sub-optimal. 2) Some failure cases listed in Haghifam et al. (2023) are due to the work of Russo & Zou (2016), on which our study is based. Improving the limitations of Russo & Zou (2016) is beyond the scope of our paper. Given that all bounds of information-theoretic methods suffer from this limitation, it is an important direction for future research.

## 7 Conclusion

This paper presents a new generalization bound for general iterative learning algorithms with bounded updates. This result is more general than previous methods, which primarily focus on the SGD algorithm. To achieve these results, we introduce a new perspective by reformulating the mutual information $I(W; S)$ as the uncertainty of the update. Our generalization bound is analyzed under various settings. Our work achieves a better vanishing rate guarantee than previous work (Neu et al., 2021) in the overparameterized regime where $d = \Theta(n)$. Finally, we examine the gap between our theory and practice by analyzing the previously discovered scaling behavior in large language models. Our model shed light on developing practial used generalization theory.

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

## A  Appendix

## B  Proof of Theorem 4.4

**Theorem B.1.** *(Theorem 1 of Xu & Raginsky (2017)) Under Assumption 4.3, the following bound holds:*

$$\left| gen(\mu, \mathbb{P}_{W|S_n}) \right| \leq \sqrt{\frac{2R^2}{n} I(W; S_n)} \tag{15}$$

**Theorem B.2.** *Under Assumption 4.3, the following bound holds:*

$$\left| gen(\mu, \mathbb{P}_{W|S_n}) \right| \leq \sqrt{\frac{2R^2}{n} [h(U^{(T)}|W_0) - h(U^{(T)}|W_0, S_n)]} \tag{16}$$

*Proof.* Using the chain rule of entropy, we have

$$h(W_T, W_0) = h(W_T|W_0) + h(W_0), \tag{17}$$

and

$$h(W_T, W_0) = h(W_0|W_T) + h(W_T). \tag{18}$$

Taking (18) - (17), we have

$$h(W_T) = h(W_T|W_0) + h(W_0) - h(W_0|W_T) \tag{19}$$

$h(W_0|W_T) + h(W_T|S_n)$ can be lower bounded by

$$
\begin{aligned}
h(W_0|W_T) + h(W_T|S_n) &\overset{(\star)}{=} h(W_0|W_T, S_n) + h(W_T|S_n) \\
&= h(W_0, W_T|S_n) \\
&= h(W_0|S_n) + h(W_T|W_0, S_n) \\
&\overset{(*)}{=} h(W_0) + h(W_T|W_0, S_n),
\end{aligned} \tag{20}
$$

where $(\star)$ and $(*)$ are due to the independent between $W_0$ and $S_n$.

$$
\begin{aligned}
I(W_T; S_n) &= h(W_T) - h(W_T|S_n) \\
&\overset{(\star)}{=} h(W_T|W_0) + h(W_0) - h(W_0|W_T) - h(W_T|S_n) \\
&\overset{\circ}{=} h(W_T|W_0) + h(W_0) - h(W_0) - h(W_T|W_0, S_n) \\
&= h(W_T|W_0) - h(W_T|W_0, S_n) \\
&= h(W_0 + U^{(T)}|W_0) - h(W_0 + U^{(T)}|W_0, S_n) \\
&= h(U^{(T)}|W_0) - h(U^{(T)}|W_0, S_n)
\end{aligned} \tag{21}
$$

where $(\star)$ is due to Equation (19) and $(\circ)$ is due to 20.

Combining with Theorem B.1, we conclude the Theorem B.2. ☐

## C  Proof of Theorem 4.5 and Theorem 4.7

**Lemma C.1.** *(From Pensia et al. (2018) page 12) If a random variable $X$ has $\mathbb{E} \|X\|^2 \leq C$, then we have $h(X) \leq \frac{d}{2} \log(\frac{2\pi eC}{d})$.*

**Lemma C.2.** *(Entropy Power inequality (Shannon, 1948)) If $X,Y$ are independent random variables with dimension $d$, then we have*

$$N(X + Y) \geq N(X) + N(Y), \tag{22}$$

*where $N(X) = \frac{1}{2\pi e} e^{\frac{2}{d} h(X)}$.*

**Definition C.3.** We say a learning algorithm is a high randomness learning algorithm if $h(U^{(T)}|W_0, S_n) \geq 0$.

**Theorem C.4.** *Under Assumption 4.3, for high randomness learning algorithm, i.e. $h(U^{(T)}|W_0, S_n) \geq 0$, the generalization error of the final iteration satisfies*

$$\left|gen(\mu, \mathbb{P}_{W|S_n})\right| \leq \sqrt{\frac{2\pi eR^2 \mathbb{V}(U^{(T)})}{n}}. \tag{23}$$

*Proof.* According to Theorem 4.4, we have

$$\left|gen(\mu, \mathbb{P}_{W|S_n})\right| \leq \sqrt{\frac{2R^2}{n}[h(U^{(T)}|W_0) - h(U^{(T)}|W_0, S_n)]} \leq \sqrt{\frac{2R^2}{n}h(U^{(T)}|W_0)}. \tag{24}$$

Using Lemma C.1

$$h(U^{(T)}|W_0) \leq \frac{d}{2} \log\left(\frac{2\pi e\mathbb{V}(U^{(T)})}{d}\right) \leq \frac{d}{2} \log\left(\frac{2\pi e\mathbb{V}(U^{(T)})}{d} + 1\right) \leq \pi e\mathbb{V}(U^{(T)}) \tag{25}$$

Combining the equations, we obtain

$$\left|gen(\mu, \mathbb{P}_{W|S_n})\right| \leq \sqrt{\frac{2R^2}{n}h(U^{(T)}|W_0)} \leq \sqrt{\frac{2\pi eR^2\mathbb{V}(U^{(T)})}{n}} \tag{26}$$

$\square$

**Theorem C.5.** *Under Assumption 4.3, for any $\sigma$, the generalization error of the final iterate satisfies*

$$\left|gen(\mu, \mathbb{P}_{W|S_n})\right| \leq \sqrt{\frac{R^2\mathbb{V}(U^{(T)})}{n\sigma^2}} + \Delta_\sigma \tag{27}$$

*Proof.*

$$\begin{aligned}
\left|gen(\mu, \mathbb{P}_{W|S_n})\right| &= \left|\mathbb{E}_{S_n \sim \mu^{\otimes n}, W \sim \mathbb{P}_{W|S_n}} \left[F_{S_n}(W) - F_\mu(W)\right]\right| \\
&\leq \left|\mathbb{E}_{S_n \sim \mu^{\otimes n}, W \sim \mathbb{P}_{W|S_n}} \left[F_{S_n}(\tilde{W}) - F_\mu(\tilde{W})\right]\right| + \Delta_\sigma,
\end{aligned} \tag{28}$$

where $\tilde{W} \triangleq W + \epsilon$, $\Delta_\sigma \triangleq |\mathbb{E}\left[(F_\mu(W_T) - F_\mu(W_T + \epsilon)) - (F_S(W_T) - F_S(W_T + \epsilon))\right]|$ and $\epsilon \sim \mathcal{N}(0, \sigma^2\mathbf{I})$.

Recall that

$$\mathbb{V}(U^{(t)}) \triangleq \mathbb{E}_{W_0 \sim \mathbb{P}_{W_0}} \mathbb{E}\left[\left\|U^{(t)} - \mathbb{E}U^{(t)}\right\|^2 |W_0\right]. \tag{29}$$

Denote $\mathbb{V}_{(w_0)}(U^{(t)}) = \mathbb{E}\left[\left\|U^{(t)} - \mathbb{E}U^{(t)}\right\|^2 |W_0\right]$. For all $w_0$, we have

$$h(U^{(T)}|W_0 = w_0) = h(U^{(T)} - \mathbb{E}U^{(t)}|W_0 = w_0) \overset{(\star)}{\leq} \frac{d}{2} \log(\frac{2\pi e\mathbb{V}_{(w_0)}(U^{(T)})}{d}) \tag{30}$$

The inequlity $(\star)$ is due to Lemma C.1. Since $\epsilon$ is independent with $U^{(T)}$, we have $\mathbb{V}_{(w_0)}(\tilde{U}^{(T)}) = \mathbb{V}_{(w_0)}(\tilde{U}^{(T)}) + \mathbb{V}(\epsilon)$. According to Lemma C.2, we have

$$\begin{aligned}
N(\tilde{U}^{(T)}|W_0, S_n) &= N(U^{(T)} + \epsilon|W_0, S_n) \\
&\geq N(U^{(T)}|W_0, S_n) + N(\epsilon) \\
&\geq N(\epsilon).
\end{aligned} \tag{31}$$

Simplify the equation, we obtain that $h(\tilde{U}^{(T)}|W_0, S_n) \geq h(\epsilon) = \frac{d}{2}\log(\frac{2\pi e \mathbb{V}(\epsilon)}{d})$.

Therefore, we have

$$
\begin{aligned}
h(\tilde{U}^{(T)}|W_0) &- h(\tilde{U}^{(T)}|W_0, S_n) \\
&\leq \int h(\tilde{U}^{(T)}|W_0)\mathrm{d}\mathbb{P}(W_0) - h(\epsilon) = \int h(\tilde{U}^{(T)}|W_0) - h(\epsilon)\mathrm{d}\mathbb{P}(W_0) \\
&\leq \int \frac{d}{2}\log(\frac{2\pi e[\mathbb{V}_{(w_0)}(U^{(T)}) + \mathbb{V}(\epsilon)]}{d}) - \frac{d}{2}\log(\frac{2\pi e \mathbb{V}(\epsilon)}{d})\mathrm{d}\mathbb{P}(W_0) \\
&= \int \frac{d}{2}\log(1 + \frac{\mathbb{V}_{(w_0)}(U^{(T)})}{\mathbb{V}(\epsilon)})\mathrm{d}\mathbb{P}(W_0) \leq \frac{d\int \mathbb{V}_{(w_0)}(U^{(T)})\mathrm{d}\mathbb{P}(W_0)}{2\mathbb{V}(\epsilon)} \\
&= \frac{\mathbb{V}(U^{(T)})}{2\sigma^2}.
\end{aligned}
\tag{32}
$$

Combining with Theorem B.2, we establish this Theorem. $\qquad\square$

## D  Proof of Theorem 4.10

**Theorem D.1.** *If the learning algorithm has bounded updates on data distribution $\mu$ and loss function $f(\cdot)$, then we have*

$$
\mathbb{V}(U^{(T)}) \leq \sum_{t=1}^{T} 4\eta_t^2 L^2 + 2L^2 \sum_{t=1}^{T} \eta_t \sum_{i=1}^{t-1} \eta_t
\tag{33}
$$

*then under Assumption 4.3, we have*

$$
gen(\mu, \mathbb{P}_{W|S_n}) \leq \sqrt{\frac{R^2}{n\sigma^2}\left(\sum_{t=1}^{T} 4\eta_t^2 L^2 + 2L^2 \sum_{t=1}^{T} \eta_t \sum_{i=1}^{t-1} \eta_t\right)} + \Delta_\sigma.
\tag{34}
$$

*If the learning algorithms have high randomness, i.e. satisfying $h(U^{(T)}|W_0, S_n) \geq 0$, we have*

$$
\left|gen(\mu, \mathbb{P}_{W|S_n})\right| \leq \sqrt{\frac{2\pi e R^2}{n}\left(\sum_{t=1}^{T} 4\eta_t^2 L^2 + 2L^2 \sum_{t=1}^{T} \eta_t \sum_{i=1}^{t-1} \eta_t\right)}.
\tag{35}
$$

*Proof.* We first derive upper bound $\mathbb{V}(U^{(T)})$ with Assumption 4.9. According to the definition, we have

$$
\begin{aligned}
\mathbb{V}(U^{(t)}) &= \mathbb{E}\left[\left\|U^{(t)} - \mathbb{E}U^{(t)}\right\|^2 |W_0\right] \\
&= \mathbb{E}\left[\left\|U^{(t-1)} - \mathbb{E}U^{(t-1)} + U_t - \mathbb{E}U_t\right\|^2 |W_0\right] \\
&= \mathbb{E}\left\|\bar{U}^{(t-1)} + \bar{U}_t\right\|^2 \\
&= \mathbb{E}\left\|\bar{U}^{(t-1)}\right\|^2 + \mathbb{E}\left\|\bar{U}_t\right\|^2 + \mathbb{E} < \bar{U}^{(t-1)}, \bar{U}_t > \\
&= \mathbb{V}(U^{(t-1)}) + \mathbb{V}(U_t) + 2\mathbb{C}(U^{(t-1)}, U_t).
\end{aligned}
\tag{36}
$$

By iterative appling Equation (36), we have:

$$
\mathbb{V}(U^{(T)}) = \sum_{t=1}^{T} \mathbb{V}(U_t) + 2\sum_{t=1}^{T} \mathbb{C}(U^{(t-1)}, U_t).
\tag{37}
$$

Due to the Assumption 4.9 and definition of $\mathbb{V}(U_t)$, we have

$$
\begin{aligned}
\mathbb{V}(U_t) &= \mathbb{E}\left\|U_t - \mathbb{E}U_t\right\|^2 \\
&\leq \left\|\eta_t L + \eta_t L\right\|_2^2 \\
&= 4\eta_t^2 L^2.
\end{aligned}
\tag{38}
$$

Then, for $U^{(t)}$, we have

$$
\left\|U^{(t)}\right\| = \left\|\sum_{i=1}^{t} U_t\right\| \leq \sum_{i=1}^{t} \left\|U_t\right\| \leq L \sum_{i=1}^{t} \eta_t
\tag{39}
$$

Therefore, we have

$$
\mathbb{C}(U^{(t)}, U_t) \leq \left\|U^{(t-1)}\right\| \left\|U_t\right\| \leq L^2 \eta_t \sum_{i=1}^{t-1} \eta_t
\tag{40}
$$

Combining the equation above, we have

$$
\mathbb{V}(U^{(T)}) = \sum_{t=1}^{T} \mathbb{V}(U_t) + 2\sum_{t=1}^{T} \mathbb{C}(U^{(t)}, U_t) \leq \sum_{t=1}^{T} 4\eta_t^2 L^2 + 2L^2 \sum_{t=1}^{T} \eta_t \sum_{i=1}^{t-1} \eta_t
\tag{41}
$$

If $\eta_t = \eta$, we have

$$
\mathbb{V}(U^{(T)}) \leq 4T\eta^2 L^2 + 2T^2\eta^2 L^2 = 2T\eta^2 L^2(2+T) = \mathcal{O}(T\eta^2(2+T)),
\tag{42}
$$

Combining with Theorem C.4, Theorem C.5, we establish this Theorem. $\qquad\square$

# E Analyzing the learning rate setting $\eta_t = c/t$

**Lemma E.1.** *(Harmonic series (Rice, 2011))* $\hbar_t \triangleq \sum_{k=1}^{t} \frac{1}{k} = \log n + \gamma + \frac{1}{2t} - \varsigma_t$, *where* $\gamma \approx 0.5772$ *and* $0 \leq \varsigma_t \leq \frac{1}{8t^2}$

**Theorem E.2.** *If the learning algorithm has bounded updates on data distribution $\mu$ and loss function $f(\cdot)$, then under Assumption 4.3, by setting $\eta_t = \frac{c}{t}$, we obtain*

$$
\mathbb{V}(U^{(T)}) = \mathcal{O}\left(c^2 \left(\log T\right)^2\right)
\tag{43}
$$

*Proof.* From the proof of Theorem D.1, we have

$$
\mathbb{V}(U^{(T)}) \leq \sum_{t=1}^{T} 4\eta_t^2 L^2 + 2L^2 \sum_{t=1}^{T} \eta_t \sum_{i=1}^{t-1} \eta_t.
\tag{44}
$$

If $\eta_t = \frac{c}{t}$, then we have

$$
\mathbb{V}(U^{(T)}) \leq 2L^2 c^2 \left(2\sum_{t=1}^{T} \frac{1}{t^2} + \sum_{t=1}^{T} \frac{1}{t} \sum_{i=1}^{t-1} \frac{1}{i}\right)
$$

$$
\begin{aligned}
&\leq 2L^2 c^2 \left(2\left(1 + \sum_{t=2}^{T} \frac{1}{t(t-1)}\right) + \sum_{t=1}^{T} \frac{1}{t} \sum_{t=1}^{T} \frac{1}{t}\right) \\
&\leq 2L^2 c^2 \left(2\left(1 + \sum_{t=2}^{T} \left(\frac{1}{t-1} - \frac{1}{t}\right)\right) + \left(\sum_{t=1}^{T} \frac{1}{t}\right)^2\right) \\
&\overset{(\star)}{\leq} 2L^2 c^2 \left(2\left(2 - \frac{1}{T}\right) + \hbar_T^2\right),
\end{aligned}
\tag{45}
$$

where $(\star)$ leverage the Lemma E.1. Obviously, we have $\hbar = \mathcal{O}(\log n)$. Therefore, we have

$$\mathbb{V}(U^{(T)}) \leq 2L^2 c^2 \left( 2 \left( 2 - \frac{1}{T} \right) + \hbar_n^2 \right) \leq 2L^2 c^2 \left( 4 + \hbar_n^2 \right) = \mathcal{O}\left( c^2 \left( \log T \right)^2 \right). \tag{46}$$

We establish the Theorem. $\qquad\square$

**Asymptotic analysis when increase** $n$    The result of the setting $\eta_t = \frac{c}{t}$ with $c = \mathcal{O}(1)$ and $T = \mathcal{O}(n)$ have a extra $\log n$ term compared with the setting where $\eta_t = \eta$ and $\eta T = \Theta(1)$. The $\log n$ is usually ignorable compared with polynominal term. Therefore, we can conclude that both settings obtain a similar results.

Table 3: Asymptotic analysis when increase $n$ under $c = \mathcal{O}(1)$ and $T = \mathcal{O}(n)$ for different situations. ($\triangle$) stands for $h(U^{(T)}|W_0, S_n) < 0$, and $(\star)$ is short for $h(U^{(T)}|W_0, S_n) \geq 0$.

| | Settings | | Ours | |
|---|---|---|---|---|
| $d$ | $\Delta_\sigma$ | | $(\triangle)$ | $(\star)$ |
| $\Theta(1)$ | $\Theta(d\sigma^2)$, $\Theta(\sigma^2/c)$ ,$\Theta(\sigma^2)$ | | $\mathcal{O}(\log n/n^{\frac{1}{3}})$ | |
| $\Theta(n)$ | $\Theta(d\sigma^2)$ | | $\mathcal{O}(\log n)$ | |
| | $\Theta(\sigma^2/c)$ | | $\mathcal{O}(\log n/n^{\frac{1}{3}})$ | |
| | $\Theta(\sigma^2)$ | | $\mathcal{O}(\log n/n^{\frac{1}{3}})$ | |

# F    Proof of Proposition 5.1

---
**Algorithm 1** Adam
---
1: **Input:**  the loss function $f(\boldsymbol{w}, z)$, the initial point $\boldsymbol{w}_1 \in \mathbb{R}^d$, the batch size $b$, learning rates $\{\eta_t\}_{t=1}^T$,  $\boldsymbol{m}_0 = 0, \boldsymbol{v}_0 = 0$, and hyperparameters $\boldsymbol{\beta} = (\boldsymbol{\beta}_1, \boldsymbol{\beta}_2)$.
2: **For** $t = 1 \to T$:
3:    Sample a mini-batch of data $B_t$ with size $b$
4:    $\nabla f_{B_t}(w_t) = \frac{1}{b} \sum_{z \in B_t} f(\boldsymbol{w}_t, z)$
5:    $\boldsymbol{m}_t \leftarrow \boldsymbol{\beta}_1 \boldsymbol{m}_{t-1} + (1 - \boldsymbol{\beta}_1) \nabla f_{B_t}(\boldsymbol{w}_t)$
6:    $\boldsymbol{v}_t \leftarrow \boldsymbol{\beta}_2 \boldsymbol{v}_{t-1} + (1 - \boldsymbol{\beta}_2) \nabla f_{B_t}(\boldsymbol{w}_t)^{\odot 2}$
7:    $\boldsymbol{w}_{t+1} \leftarrow \boldsymbol{w}_t - \eta_t \frac{\boldsymbol{m}_t}{\sqrt{\boldsymbol{v}_t}}$
8: **End For**

---
**Algorithm 2** Adagrad
---
1: **Input:** the loss function $f(\boldsymbol{w}, z)$, the initial point $\boldsymbol{w}_1 \in \mathbb{R}^d$, the batch size $b$, learning rates $\{\eta_t\}_{t=1}^T$, $\boldsymbol{m}_0 = 0, \boldsymbol{v}_0 = 0$..
2:
3: **For** $t = 1 \to T$:
4:    Sample a mini-batch of data $B_t$ with size $b$
5:    $\nabla f_{B_t}(w_t) = \frac{1}{b} \sum_{z \in B_t} f(\boldsymbol{w}_t, z)$
6:    $\boldsymbol{v}_t \leftarrow \boldsymbol{v}_{t-1} + \nabla f_{B_t}(\boldsymbol{w}_t)^{\odot 2}$
7:    $\boldsymbol{w}_{t+1} \leftarrow \boldsymbol{w}_t - \eta_t \frac{\nabla f_{B_t}(\boldsymbol{w})}{\sqrt{\boldsymbol{v}_t}}$
8: **End For**
9:

**Lemma F.1.** *For Adam, we have* $\forall t \geq 1$, $|\boldsymbol{w}_{t+1,l} - \boldsymbol{w}_{t,l}| \leq \eta_t \frac{1 - \boldsymbol{\beta}_1}{\sqrt{1 - \boldsymbol{\beta}_2}\sqrt{1 - \frac{\beta_1^2}{\beta_2}}}$ *and for Adagrad, we have*

$\forall t \geq 1$, $|\boldsymbol{w}_{t+1,l} - \boldsymbol{w}_{t,l}| \leq \eta_t$.

*Proof.* For Adam, We have that

$$|\boldsymbol{w}_{t+1,l} - \boldsymbol{w}_{t,l}| = \eta_t \left| \frac{\boldsymbol{m}_{t,l}}{\sqrt{\boldsymbol{v}_{t,l}}} \right| \leq \eta_t \frac{\sum_{i=0}^{t-1}(1 - \boldsymbol{\beta}_1)\boldsymbol{\beta}_1^i |\boldsymbol{g}_{t-i,l}|}{\sqrt{\sum_{i=0}^{t-1}(1 - \boldsymbol{\beta}_2)\boldsymbol{\beta}_2^i |\boldsymbol{g}_{t-i,l}|^2 + \boldsymbol{\beta}_2^t \boldsymbol{v}_{0,l}}}$$

$$\leq \eta_t \frac{1 - \boldsymbol{\beta}_1}{\sqrt{1 - \boldsymbol{\beta}_2}} \frac{\sqrt{\sum_{i=0}^{t-1} \boldsymbol{\beta}_2^i |\boldsymbol{g}_{t-i,l}|^2} \sqrt{\sum_{i=0}^{t-1} \frac{\beta_1^{2i}}{\beta_2^i}}}{\sqrt{\sum_{i=0}^{t-1} \boldsymbol{\beta}_2^i |\boldsymbol{g}_{t-i,l}|^2}} \leq \eta_t \frac{1 - \boldsymbol{\beta}_1}{\sqrt{1 - \boldsymbol{\beta}_2}\sqrt{1 - \frac{\beta_1^2}{\beta_2}}}.$$

Here the second inequality is due to Cauchy's inequality. The proof is completed.

For Adagrad, we have that

$$|\boldsymbol{w}_{t+1,l} - \boldsymbol{w}_{t,l}| = \eta_t \left| \frac{g_{t,l}}{\sqrt{\sum_{i=1}^{t} g_{i,l}^2}} \right| \leq \eta_t \left| \frac{g_{t,l}}{\sqrt{g_{i,l}^2}} \right| \leq \eta_t \tag{47}$$

□

**Proposition F.2.** *Adam, Adagrad, RMSprop are bounded updates with respect to all data distribution and function $f(\cdot)$ when $d = \Theta(1)$*

*Proof.* For Adam, we have

$$\|u_t\| = \frac{1}{\eta_t} \|U_t\| \sqrt{\sum_{l=1}^{d} (\boldsymbol{w}_{t+1,l} - \boldsymbol{w}_{t,l})^2} \leq d \frac{1 - \boldsymbol{\beta}_1}{\sqrt{1 - \boldsymbol{\beta}_2} \sqrt{1 - \frac{\boldsymbol{\beta}_1^2}{\boldsymbol{\beta}_2}}} \tag{48}$$

Because RMSgrad is a special case of Adam by setting $\boldsymbol{\beta}_1 = 0$, we have

$$\|u_t\| = \leq d \frac{1}{\sqrt{1 - \boldsymbol{\beta}_2}} \tag{49}$$

For Adagrad, we have

$$\|u_t\| = \frac{1}{\eta_t} \|U_t\| \sqrt{\sum_{l=1}^{d} (\boldsymbol{w}_{t+1,l} - \boldsymbol{w}_{t,l})^2} \leq d \tag{50}$$

When $d = \mathcal{O}(1)$, all the learning algorithms can be bounded by a constant. We establish the proposition. □

**Under the setting $\eta_t = \eta$ and $\eta T = \mathcal{O}(1)$, we have $\left| gen(\mu, \mathbb{P}_{W|S_n}) \right| = \mathcal{O}(1/\sqrt{n})$.**

## G From Pac-Bayes perspective

In this section, we provided a analysis of generalization bound with update variance from pac-bayes perspective. From this perspective, we further enhance Theorem 4.7 by achieving the following improvements: **1) Removing the sub-Guassian Assumption**, and **2) Acquiring high probability bounds.**

**Theorem G.1.** *(From McAllester (1999); Dziugaite & Roy (2017)) For any prior $\mathbb{P}_1$ over parameters with probability $1 - \delta$ over the choice of the training set $S_n \sim \mu^{\otimes n}$, for any posterior $\mathbb{P}_2$, we have*

$$\mathbb{E}_{w \sim \mathbb{P}_2}(F_\mu(w)) - \mathbb{E}_{w \sim \mathbb{P}_2}(F_{S_n}(w)) \leq \sqrt{\frac{KL(\mathbb{P}_1 \| \mathbb{P}_2) + \log \frac{n}{\delta}}{2(n - 2)}} \tag{51}$$

**Theorem G.2.** *(Using Pac-Bayes Methods) for relative small $\sigma$, with probability $1 - \delta$ over the choice of the training set $S_n \sim \mu^{\otimes n}$, the generalization error of the final iteration satisfies*

$$\mathbb{E}_{w \sim \mathbb{P}_2}(F_\mu(w)) - \mathbb{E}_{w \sim \mathbb{P}_2}(F_{S_n}(w)) \leq \sqrt{\frac{\mathbb{V}(W_0) + \mathbb{V}'(U^{(T)}) + \sigma^2 \log \frac{n}{\delta}}{2\sigma^2(n - 2)}} + \Delta_\sigma, \tag{52}$$

*where $\mathbb{V}(W_0) = \mathbb{E} \|W_0 - \mathbb{E}W_0\|^2$ and $\mathbb{V}'(U^{(T)}) = \mathbb{E} \|U^{(T)} - \mathbb{E}U^{(T)}\|^2$.*

*Remark* G.3. $\mathbb{V}'(U^{(T)})$ is different from $\mathbb{V}(U^{(T)})$ in considering the randomness of initialization weights $W_0$. By keeping the $W_0$ as a constant value, we achieve the outcomes of Theorem 4.7, except that the sub-Gaussian assumption is removed and high probability bounds are presented.

*Proof.* If $\mathbb{P}_1 = \mathcal{N}(\boldsymbol{\mu}_p, \sigma_p^2 \mathbf{I})$ and $\mathbb{P}_2 = \mathcal{N}(\boldsymbol{\mu}_q, \sigma_q^2 \mathbf{I})$, then we have

$$KL(\mathbb{P}_1 \| \mathbb{P}_2) = \frac{1}{2} \left[ \frac{k\sigma_q^2 + \|\boldsymbol{\mu}_p - \boldsymbol{\mu}_q\|^2}{\sigma_p^2} - k + k \log \left( \frac{\sigma_p^2}{\sigma_q^2} \right) \right] \tag{53}$$

If $\sigma_p = \sigma_q$, then we have

$$KL(\mathbb{P}_1 \| \mathbb{P}_2) = \frac{\|\boldsymbol{\mu}_p - \boldsymbol{\mu}_q\|^2}{2\sigma_p^2}. \tag{54}$$

By setting the $\mathbb{P}_1 = \mathcal{N}(\boldsymbol{\mu}_p, \sigma_p^2 \mathbf{I})$ and for each $W_T$, setting $P_2 = \mathcal{N}(W_T, \sigma_q^2 \mathbf{I})$, and $\sigma_p = \sigma_q = \sigma$, combining with Theorem G.1, we obtain that

$$\mathbb{E}_{w \sim \mathbb{P}_2}(F_\mu(w)) - \mathbb{E}_{w \sim \mathbb{P}_2}(F_{S_n}(w)) \leq \sqrt{\frac{\mathbb{E}\|W_T - \boldsymbol{\mu}_p\|^2 + 2\sigma^2 \log \frac{n}{\delta}}{4\sigma^2(n-2)}} + \Delta_\sigma. \tag{55}$$

Then, in the following we use distribution dependent prior to obtain a update uncertainty term. We set $\boldsymbol{u}_p = \mathbb{E}W_T$, because the expectation is taken from the randomness of algorithm and the randomness of sampling, $\mathbb{E}W_T$ **is distribution dependent but training data independent**. Then, we have

$$\begin{aligned} \mathbb{E}\|W_T - \boldsymbol{\mu}_p\|^2 &= \mathbb{E}\left\|W_0 + U^{(T)} - \mathbb{E}W_0 - \mathbb{E}U^{(T)}\right\|^2 \\ &\leq 2\mathbb{E}\|W_0 - \mathbb{E}W_0\|^2 + 2\mathbb{E}\left\|U^{(T)} - \mathbb{E}U^{(T)}\right\|^2 \\ &= 2\mathbb{V}(W_0) + 2\mathbb{V}'(U^{(T)}) \end{aligned} \tag{56}$$

Therefore, we have

$$\mathbb{E}_{w \sim \mathbb{P}_2}(F_\mu(w)) - \mathbb{E}_{w \sim \mathbb{P}_2}(F_{S_n}(w)) \leq \sqrt{\frac{\mathbb{V}(W_0) + \mathbb{V}'(U^{(T)}) + \sigma^2 \log \frac{n}{\delta}}{2\sigma^2(n-2)}} + \Delta_\sigma. \tag{57}$$

$\square$

## H  Discussion of the uncertainty of update with other measures

**Compared to variance of gradient**  A closely related measure is the variance of the gradient. Given the dataset $S_n = \{z_i\}_{i=1}^n$, the variance of the gradient in weights $w$ is calculated as $\frac{1}{n}\sum_{i=1}^n \|\nabla f(W, z_i) - F_S(W)\|$. The generalization bounds of Fu et al. (2023); Neu et al. (2021) depend on this type of variance. The key distinction is that the update certainty is a joint property of the learning algorithm and the function $f(\cdot)$. Another difference is that the update uncertainty is evaluated by sampling different $S_n \sim \mu^{\otimes n}$, whereas the variance of the gradient is assessed using only $S_n$.

**Connection to stability measure**  Another most used technique for analyzing the generalization behavior is analyzing the stability behavior of learning algorithm (Hardt et al., 2016; Lei & Ying, 2020; Liu et al., 2017; Bassily et al., 2020; Feldman & Vondrak, 2019). We analyze the connection between on-average stabilty(Lei & Ying, 2020) and the variance of update. The connection between on-average stability and uniform stability is discussed in Lei & Ying (2020). Given two set $S_n = \{z_i\}_{i=1}^n$ and $\tilde{S}_n = \{\tilde{z}_i\}_{i=1}^n$. $S^{(i)} \triangleq \{z_1 \cdots z_{i-1}, \tilde{z}_i, z_{i+1} \cdots z_n\}$ The $l_2$ on-average stable is calculated as

$$\mathbb{E}_{S_n, \tilde{S}_n, \mathcal{A}} \left[ \frac{1}{n} \sum_{i=1}^n \left\| \mathcal{A}(S_n) - \mathcal{A}(S_n^{(i)}) \right\| \right] \leq \mathrm{stab} \tag{58}$$

Then, we define $S_n^{[k]}$ set, where $k$ samples of $S_n^{[k]}$ come from $S_n$ and the others come from $S_n'$. We denote $\mathbb{E}_k \mathcal{A} = \mathbb{E}_k \mathcal{A}(S_n^{[k]})$, where the expectation is taken regarding the randomness of sampling of $S_n'$,

the randomness of chosing $k$ samples from $S_n$, as well as the randomness of learning algorithm $\mathcal{A}$. If $\mathbb{E}\left[\left\|\mathbb{E}_k\mathcal{A} - \mathbb{E}_{k-1}\mathcal{A}\right\|^2 |W_0\right] \approx \mathbb{E}\left[\left\|\mathbb{E}_i\mathcal{A} - \mathbb{E}_{i-1}\mathcal{A}\right\|^2 |W_0\right]$ for all $k, i \in [n]$, we have

$$
\begin{aligned}
\mathbb{V}(U^{(T)}) &= \mathbb{E}\left[\left\|U^{(T)}|S_n - \mathbb{E}U^{(T)}\right\|^2 |W_0\right] = \mathbb{E}\left[\left\|W^{(T)}|S_n - \mathbb{E}W^{(T)}\right\|^2 |W_0\right] \\
&= \mathbb{E}\left[\left\|\mathcal{A}(S_n) - \mathbb{E}\mathcal{A}(S_n')\right\|^2 |W_0\right] \le \mathbb{E}\left[\sum_{k=1}^n \left\|\mathbb{E}_k\mathcal{A} - \mathbb{E}_{k-1}\mathcal{A}\right\|^2 |W_0\right] \\
&\approx \mathbb{E}\left[n\left\|\mathbb{E}_n\mathcal{A} - \mathbb{E}_{n-1}\mathcal{A}\right\|^2 |W_0\right] \approx n\mathbb{E}_{S_n, \tilde{S}_n, \mathcal{A}}\left[\frac{1}{n}\sum_{i=1}^n \left\|\mathcal{A}(S_n) - \mathcal{A}(S_n^{(i)})\right\|\right]
\end{aligned}
\tag{59}
$$

Therefore, we have $\mathbb{V}(U^{(T)}) \le n\,\text{Stab}$.

