# OpenReview forum: "Generalization error bounds for iterative learning algorithms with bounded updates"
_TMLR — Withdrawn by Authors_

### Review · Reviewer_naVm · 2024-11-11

**Summary Of Contributions:**

This submission proves bounds on the generalization error of learning algorithms in terms of certain mutual information quantities. I will outline the main results. Let $S$ be a dataset, $W_0$ the randomized initialization,  $W$ the final model, and $U = W - W_0$ the "model update."
- Theorem 4.4 bounds the generalization error in terms of $I(U; S\mid W_0)$, i.e., the information about the update. This is a restatement of the results Russo and Zou and Xu and Raginsky, which are in terms of $I(W;S)$. The proof, as I understand it, shows that these two terms are equal.
- Theorem 4.5 is a corollary of Theorem 4.4, relating entropy to variance to give a bound in terms of $\mathbb{E}_{W_0}[\mathrm{Var}(U)]$. The proof uses the fact that Gaussians maximize entropy subject to a variance constraint.
- Theorem 4.7 analyzes a "surrogate process" similar to prior work. We consider, as a thought experiment, a perturbed vector $\widetilde{W} = W + N(0,\sigma^2\mathbb{I})$. While $\widetilde{W}$ will have different generalization behavior than $W$, the noise allows us to apply the variance bound to it.
- Theorem 4.10 bounds the variance of $U$ by assuming that the algorithm is iterative and each update is bounded.

**Audience:**

No

**Broader Impact Concerns:**

None.

**Claims And Evidence:**

No

**Requested Changes:**

Before I recommend acceptance, this submission would need a significantly improved presentation, clearer comparison with prior work, and a *very* clear articulation of how its results improve our understanding of generalization.

**Strengths And Weaknesses:**

This paper considers a topic of interest to the TMLR community. The statements here are, to the best of my knowledge, new. However, I do not understand what the results show us above and beyond the papers of Russo and Zou (2016), Xu and Raginsky (2017), Pensia et al. (2018), and Neu et al. (2021). I do not agree, as the submission claims, that this work "represents a significant step towards developing practically useful generalization theories."

One of the submission's main claimed novelties is in Theorem 4.4, which reformulates the bound of Xu and Raginsky in terms of "update uncertainty." Here's a proof of that theorem. XR give a bound in terms of $I(W;S)$. We have a Markov chain $W_0 \leftrightarrow W \leftrightarrow S$ and $W_0\perp S$, so $I(W;S) = I(W,W_0;S) = I(W;S\mid W_0)$. Since $W = W_0 + U$, we arrive at $I(W;S) = I(U; S\mid W_0)$.

Another of the submission's claimed contributions is to extend existing analysis to algorithms other than SGD. But I understand the analysis to be a "one-step" version of Pensia et al.: they seem to analyze bounded updates followed by noise. The connections between bounded updates, bounded variance, and entropy are explicit in that paper.

The submission also compares to Neu et al., about which I have some questions. The submission says "Neu et al. adds noise throughout the entire process." (p7). But their Section 4 says "we define a peturbed version of the output as $\widetilde W_T = W_T + \xi_t$, where... $\xi_T\sim \mathcal{N}(0, \sigma_{1:T}^2 I)$". It seems to me they use the same surrogate process with a more involved analysis. Thus, I am confused by many of the conclusions drawn in the submission's Section 5.3, "Comparison with Neu et al. (2021)".

Additionally, I found the presentation rather poor. Here are a few issues.
- The submission defines an algorithm to be "high randomness" if $h(U \mid W_0,S) >0$, using my simplified notation. But I don't understand: [Wikipedia](https://en.wikipedia.org/wiki/Differential_entropy#Properties_of_differential_entropy) tells me that $h(a X)= h(X) + \log|a|$, so it seems I can make any algorithm "high randomness" by rescaling its parameter space.
- The main theorems' proofs are simple (which is fine!) but are pushed to appendices and not written well. The authors should work to simplify their presentation, highlighting the key steps, and put the proofs in the main text.
- Poor language and editing: many instances of "Guassian"; Eq. (5) should be an outer product, not an inner product; after (7) the sentence "The update entropy can be interprete as of measure the uncertainty."

---

### Review · Reviewer_ztae · 2024-11-12

**Summary Of Contributions:**

This paper studies the generalization characteristics of iterative learning algorithms with bounded updates, using information-theoretic approaches. The study derives a novel generalization error bound by reformulating mutual information as an uncertainty measure of model updates. The authors introduce a variance decomposition technique, in order to remove the reliance on the conventional chaining rule of mutual information. This innovation yields improved generalization bounds in various contexts. The authors also validate the generalization behavior of these bounds across different types of bounded-update learning algorithms, including Adam, Adagrad, and RMSprop.

**Audience:**

Yes

**Claims And Evidence:**

No

**Requested Changes:**

- The description of the theoretical results needs to be further revised, as there are several typos and imprecise statement.
- For example: on Page 2, what does the notation $Q << P$ mean? Notations $f(W, Z), f(w, Z), f(W, z)$ are also used interchangeably. it would be better to keep the notation consistent. What does the $h()$ in Theorem 4.4 mean?
- In Theorem 4.5, it states a "high randomness learning algorithm". It would be better to provide a precise definition for this.


- It would be better to provide a preliminary of the existing bounds before Section 4. It would help to understand the contribution of this paper.

- In Section 4.2, this work builds the results on learning algorithms with bounded updates. It would be better to show empirical observations on Adam, Adagrad, and RMSprop to show that they have bounded updates. What are their $L$ values?

- Table 1 is hard to interpret. It would be better to reconstruct the table more.

**Strengths And Weaknesses:**

### Strengths:

- The reformulation of mutual information to express uncertainty in updates is a valuable perspective shift. This work also employes a variance decomposition instead of the chaining rule, to derive a surrogate process for analyzing generalization in iterative algorithms.
- The paper compares its results with previous works, such as Neu et al. (2021), and achieves improvements in the vanishing rate guarantee of generalization error bounds.

### Weaknesses:

- The improvement over the existing generalization bounds are not clearly stated. For example, the authors provide a comparison with existing works in Table 2. However, it is not clear where the factor of $log n$ is derived from the bound in Equation 12. Moreover, what the typical values of the constants in the bound, for example $R$ and $L$.

- It would be better to provide empirical findings to validate the theoretical results and validate the assumptions. For example, what would the constants in the bound ($R$ and $L$) be in applications? How well does the practical evaluation of the bound correlate the empirical generalization errors.

---

### Review · Reviewer_bxeh · 2024-11-19

**Summary Of Contributions:**

This paper offers a novel viewpoint for analyzing the mutual information $I(W,S_n)$ by the contribution dataset $S_n$ to the update uncertainty. This later enables adding noise only to the final iteration during the surrogate process, which exhibits weaker dependence on T comparing to information-theoretic method in Neu et al. (2021).

**Audience:**

Yes

**Claims And Evidence:**

Yes

**Requested Changes:**

- Compare the derived generalization error bound to information-theoretic method Neu et al. (2021) without fixing $\eta T = \Theta (1)$ in their work, and takes d into account.
- Compare the derived general generalization error bound to stability-based work in mini-batch setting. For example, Stability and Generalization for Minibatch SGD and Local SGD by Yunwen Lei, Tao Sun, Mingrui Liu, arxiv 2023.

**Strengths And Weaknesses:**

Strengths:
- Besides the contributions discussed above, the theory can apply to general learning algorithms that satisfy the bounded updates assumption.

Weaknesses:
- The discussion on the overparameterized scenario is not complete, see requested changes.
- Missing discussion with literature, see requested changes.

---

### Note · Authors · 2024-12-09

**Comment:**

We would like to express our sincere gratitude to the reviewers for their valuable feedback. Their comments are instrumental in improving the quality of the paper. Unfortunately, due to time constraints, we are unable to address all of the raised concerns at this stage. As a result, we have decided to withdraw the paper for now. We will continue revising it, taking the reviewers' suggestions into account, and plan to submit it for future publication.

**Withdrawal Confirmation:**

I have read and agree with the venue's withdrawal policy on behalf of myself and my co-authors.